neuroscience/mathematical modelling

anaesthesia, consciousness, ketamine, propofol, attractor, dynamics

**Author for correspondence:**
Thomas F. Varley
e-mail: tvarley@iu.edu

# Topological analysis of differential effects of ketamine and propofol anaesthesia on brain dynamics

Thomas F. Varley[1,2], Vanessa Denny[1], Olaf Sporns[1,3] and Alice Patania[3]

[1]Psychological & Brain Sciences, and [2]School of Informatics, Computing and Engineering, Indiana University, Bloomington, IN 47401, USA
[3]Indiana University Network Sciences Institute (IUNI), Bloomington, IN 47401, USA

 TFV, 0000-0002-3317-9882; AP, 0000-0002-3047-4376

Research has found that the vividness of conscious experience is related to brain dynamics. Despite both being anaesthetics, propofol and ketamine produce different subjective states: we explore the different effects of these two anaesthetics on the structure of dynamic attractors reconstructed from electrophysiological activity recorded from cerebral cortex of two macaques. We used two methods: the first embeds the recordings in a continuous high-dimensional manifold on which we use topological data analysis to infer the presence of higher-order dynamics. The second reconstruction, an ordinal partition network embedding, allows us to create a discrete state-transition network, which is amenable to information-theoretic analysis and contains rich information about state-transition dynamics. We find that the awake condition generally had the 'richest' structure, visiting the most states, the presence of pronounced higher-order structures, and the least deterministic dynamics. By contrast, the propofol condition had the most dissimilar dynamics, transitioning to a more impoverished, constrained, low-structure regime. The ketamine condition, interestingly, seemed to combine aspects of both: while it was generally less complex than the awake condition, it remained well above propofol in almost all measures. These results provide deeper and more comprehensive insights than what is typically gained by using point-measures of complexity.

## 1. Introduction

In recent decades, the study of neural correlates of consciousness has developed into a rich and rapidly maturing field of research. A core component of the study of consciousness is the use of

consciousness-altering drugs, which provide a mapping between measurable differences in brain dynamics and specific qualities of conscious experience [1]. Pharmacology has revealed the molecular actions of different drugs [2], and recent research has begun to look at comparing brain dynamics induced by different drugs with a specific focus on how different dynamics might relate to conscious awareness [3,4]. In this paper, we take such a comparative approach to explore the differences between the effects of propofol and ketamine on multi-scale brain dynamics with an eye specifically to how these dynamics might explain the differences in consciousness induced by both drugs. Here, we characterize brain dynamics by adapting two complementary models capturing the evolution of whole-brain states through time: the first plots a trajectory through a high-dimensional configuration space, while the other discretizes transitions into a Markovian state-transition network.

While ketamine and propofol are both classified broadly as anaesthetics, and both obliterate consciousness at high doses, they are useful to compare due to their markedly different pharmacologies and the differences between the states they induce at low-to-moderate doses. Propofol is one of the most commonly-used anaesthetics in medicine, and while its full mechanism of action is not totally understood, a key feature is believed to be its widespread modulation of $GABA_A$ receptors [2,5]. By binding to $GABA_A$rs, propofol potentiates the effects of endogenous GABA, causing widespread inhibition of neuronal activity. Consequently, even at low doses, propofol induces states of amnesia, sedation and atonia, and at higher doses, full anaesthesia. Propofol reliably induces a suit of changes to oscillatory activity in the brain, such as the emergence of widespread frontal alpha patterns [6], as well as regionally specific changes to gamma and beta bands indicative of a multi-stage fragmenting of neural communication networks [7].

In contrast to propofol, ketamine acts primarily as an antagonist of glutamaterigic NMDA receptors, which has a local excitatory response [2,8]. Blockade of NMDA receptors has been found to disinhibit activity in cortical pyramidal neurons, driving activity in local cortical cirtuits [9]. Ketamine causes widespread, weak central nervous system stimulation, in contrast to propofol's deeply sedating properties. Despite this increase in activity, ketamine has been found to significantly disrupt directed information flow across the cortex [10] in a manner consistent with the network fragmentation observed under GABAergic surgical anaesthetics [11,12]. Like propofol, ketamine is associated with a distinct pattern of changes to oscillatory power with a particular focus on the emergence of high-frequency activity in the 20–70 Hz band, which distinguishes it from other commonly used surgical anaesthetics [13] and suggests that the different pharmacologies of the two drugs result in different 'paths to unconsciousness,' described by different dynamical regimes [14]. The state that ketamine induces is typically referred to as 'dissociative anaesthesia' and represents a highly atypical state of consciousness [15,16]. In a state of dissociative anaesthesia, an individual will often appear to be unresponsive to stimuli (including pain) and, to an external observer, may be indistinguishable from someone anaesthetized with a typical anaesthetic like propofol. Unlike propofol, which simply ablates consciousness, an otherwise unresponsive patient anaesthetized with ketamine often continues to have complex, conscious experiences, including hallucinations, out-of-body experiences, and dream-like, immersive experiences [8].

It has been hypothesized that the differences between ketamine and propofol anaesthesia are the result of changes to the global brain dynamics induced by each drug [4]. Experimental evidence suggests propofol inhibits the ability of the brain to maintain high levels of dynamical complexity, resulting in a low-entropy state insufficient for supporting conscious awareness [17,18]. By contrast, ketamine's blockade of NMDArs disinhibits cortical neurons, causing widespread, uncoordinated excitatory activity [3,4,8,9]. This may result in an increase in the entropy of brain activity without abolishing consciousness, artificially expanding (or at least altering) the state-space repertoire. The hypothesis that a dynamic state of higher-than-normal entropy might correspond to a psychedelic or hallucinatory state of consciousness has become known as the entropic brain hypothesis [19,20] and received empirical support from studies of psychedelic drugs [21]. The majority of these studies rely ultimately on point-summaries of 'complexity' (e.g. Lempel–Ziv complexity [17,18], entropy [22,23], etc.). However, these point-summary measures, while informative, collapse multi-scale dynamics into a single number and thus have difficulty capturing its specific shape or form. For visualization of how an OPN is constructed from brain data see figure 1.

We report results relating brain dynamics to states of consciousness (normal wakefulness, ketamine anaesthesia and propofol anaesthesia) using freely available electrophysiological data from the Neurotycho project [24,25]. In the first part, we describe the structure of brain activity in terms of its evolution through a high-dimensional state-space: at every moment $t$, the system's 'state' can be described as a vector embedded in a $k$-dimensional space, one dimension for each channel. Conveniently, this embedded point cloud (EPC) does not require any dimensionality reduction, and,

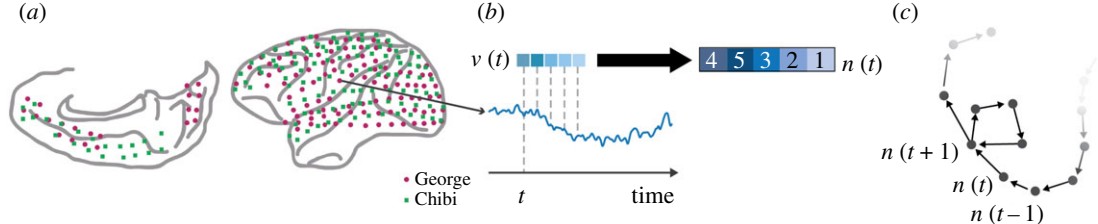

**Figure 1.** (a) An OPN was computed for each channel separately. Each channel time series is embedded in a $d$-dimensional space, using a time-lag $\tau$, as is done when constructing Taken's embedding into the phase space. The result is a temporally ordered set of vectors of length $d$, $v_t$. Each vector $v_t$ is then mapped to the permutation $\pi$, which sorts the coefficients of the vector in increasing order (b). This new vector $n_t = \pi(v_t)$ will be represented by a permutation of the numbers $1, 2, \ldots, d$. We can consider these permutations as nodes in a directed network and connect with a directed edge two permutations that come from consecutive time points (c).

conceivably, all other metrics can be reconstructed from this high-dimensional structure. One of the simplest measures is the 'distance' between successive points, from which we can derive a 'velocity', the distance traversed between subsequent time-steps, as well as reconstruct how 'far' in state-space the system moves over the course of the recording. In addition to these local measures, modelling continuous brain dynamics as a manifold renders it amenable to techniques from algebraic topology (a branch of mathematics dealing with the structures of surfaces and manifolds, particularly focused on homology). Topological data analysis (TDA) [26,27] allows us to understand details of the trajectory, including the emergence of cycles and what regions it may preferentially visit. TDA has been used extensively to characterize the chaoticity of time series [28–30] and provides a suite of techniques for classifying high-dimensional structures.

One limitation of the EPC approach is that the data are represented by continuous variables, and so every 'state' is unique, making it difficult to understand higher-level state-transition dynamics. To address this issue, in the second part of the study we discretize the EPC by creating ordinal partition networks [31–33] (OPNs) which map sets of multiple unique states to the same set of nodes in a network, and for which the probability of the system transitioning from Set A to Set B is recorded as a weighted directed edge from node A to node B. In this way, the OPN represents a reconstructed attractor that is discrete rather than continuous and consequently amenable to a number of analyses specific to discrete manifolds like networks. In particular, network science provides tools that allow us to explore the attractor at different levels: the micro-scale (node-level differences), meso-scale (community-level differences) and macro-scales (global topological differences). In combination, the two parts of our study allow a much richer understanding of brain dynamics than single-point measures can provide.

We should note that in this project we have focused primarily on the issue of *level of consciousness* rather than the *content of consciousness.* This is a subtle distinction that has been discussed in detail (for review, see [34]) but briefly, the level of consciousness quantifies the 'amount' of consciousness, such as the vividness or intensity of subjective experience, while the content of consciousness refers to the specific perceptions that are being consciously experienced. The question of the content of consciousness is well explored by psychophysical studies [35], and more recently discussed theoretically in the context of integrated information theory (IIT) [36,37]; however, it is beyond the focus of the results presented here, for several reasons. Primarily, anaesthetic states are typically light on complex contents, and macaques are unable to report their subjective experience, we have no access to the contents of their consciousness, only their status as awake or anaesthetized based on externally observable variables, and the drug in question.

# 2. Material and methods

## 2.1. Data

We used the Neurotycho dataset, an open-access collection of multidimensional, invasive electrocorticographical recordings from multiple macaque monkeys [24]. Specifically, we used data from two monkeys (Chibi and George) prior to and during behavioural unresponsiveness with

propofol or ketamine [25]. ECoG data were recorded on 128-channel, invasive recording array at a sampling rate of 1000 Hz. For both monkeys, the array covered the entire left cortical hemisphere, including the medial wall. The Neurotycho anaesthesia datasets used here have been previously used in a large number of studies assessing how drug-induced loss of consciousness alters brain dynamics (for a sample, see [38–41]).

### 2.1.1. Anaesthetic induction

The details of anaesthetic induction are described in [25], and can be viewed on the relevant Neurotycho wiki page.[1] Briefly, in the awake condition, the monkeys were restrained in a primate chair with arms and legs fixed, and neural data were collected while the monkey was calm. During the ketamine anaesthesia condition, the restrained monkeys were injected with intra-muscular ketamine (4.3 mg kg$^{-1}$ for Chibi, 5.9 mg kg$^{-1}$ for George), and anaesthesia was determined as the point that the monkey no longer responded to physical stimulus (manipulation of the hand and/or tickling the nose with a cotton swab) and slow-wave oscillations were observed in recorded data. The data were then recorded for 10 min (no supplemental or maintenance doses were given). For the propofol condition, both restrained monkeys were injected with intra-venous propofol (5.2 mg kg$^{-1}$ for Chibi, 5 mg kg$^{-1}$ for George), and loss-of-consciousness assessed using the same criteria. Recordings were then carried out for 10 min.

For each drug condition, recordings of normal consciousness were made prior to infusion of anaesthesia, and each experiment was done twice (e.g. two experiments where Chibi is anaesthetized with propofol, two with ketamine, resulting in four recordings of Chibi awake, and two of Chibi under each condition). We removed channels with intractable artefacts from the scans (four channels from Chibi, two from George). This meant that the number of channels in the final analysed data differed between monkey (124 channels for Chibi and 126 channels for George). From each recording, we manually selected six, artefact-free 10 s intervals from the anaesthetized condition and three artefact-free 10 s intervals from the awake condition (because each anaesthesia condition had its own awake condition, we had twice as much awake data, and consequently halved the number of sections taken from each scan to ensure that the total number of awake intervals matched the total number of anaesthetized intervals for each drug condition). Ten-second intervals were the largest that we found computationally tractable: the run-time and memory requirements of the topological data analysis grows super-exponentially with the number of samples, and consequently this required us to keep them comparatively short. All selections were free of major artefacts. This resulted in 24 samples each for the awake condition, the ketamine condition, and the propofol condition. The time series were visualized and manually selected in EEGLab [42].

### 2.1.2. Pre-processing

After subsections of the time series had been selected, pre-processing was performed in MNE-Python [43,44]. Each series was high-pass filtered with a low frequency of 0.1 Hz, low-pass filtered with a high frequency of 200 Hz and notch-filtered at 50 Hz and all subsequent harmonics up to 250 Hz to account for electrical line noise in Japan, we removed 50 ms from the start and the end of each sample due to filtering artefacts. All filters were FIR type and were applied forwards and backwards to avoid phase-shifting the data [45]. Following the filtering, time series from each sample were z-scored as a set (to relative distances between each moment).

## 2.2. Embedded point cloud

The analysis explored the distribution of fine-scale global activity patterns that appear over the course of the recording. In our framework, global activation patterns recorded by the array are represented as moving states in a high-dimensional configuration space. Studying the relative position and dynamics of these states gives unique insights in quantifying the underlying dynamical structure.

Each time segment can be represented as a $C \times T$ matrix $M$, where $C$ is the number of channels (126 or 124), and $T$ is the number of samples taken over the course of the recording (in this case, 9900). Each of the $T$ column-vectors in $M$ (each $C$ entries long) represents a distribution of voltages across the recording array, at each moment in time.

[1]http://wiki.neurotycho.org/Anesthesia_and_Sleep_Task_Details.

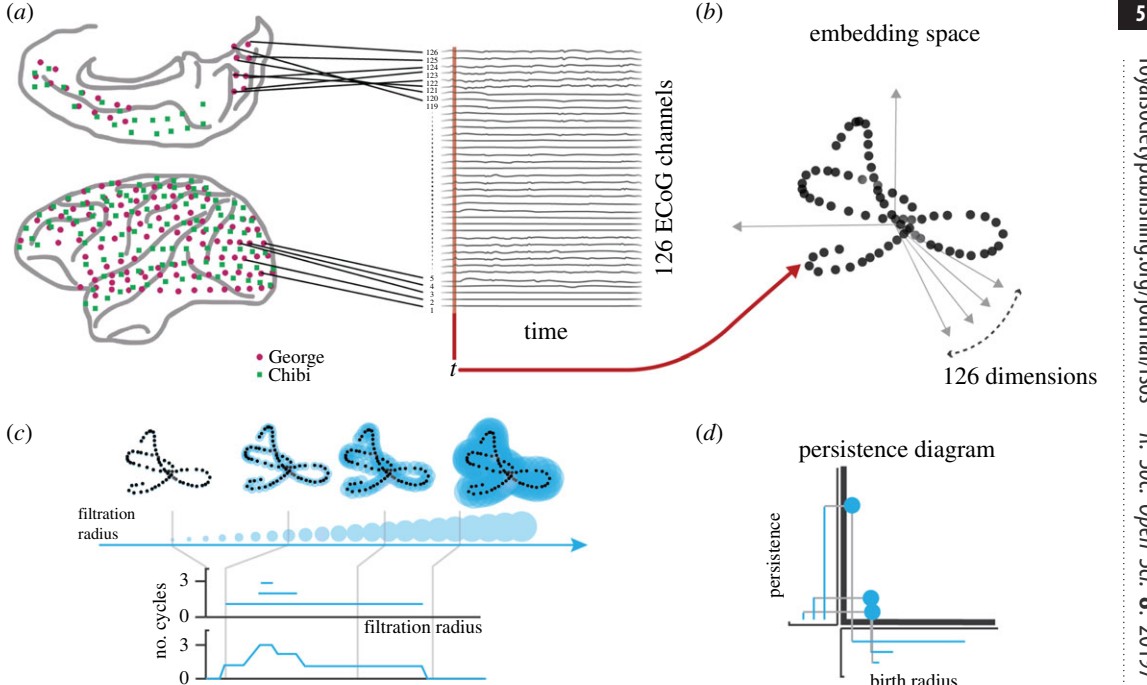

**Figure 2.** Construction of the embedded point cloud from a multidimensional time series. Every time-step in the recording corresponds to a single column vector in a $C \times T$ array (*a*), and this vector can be imagined as a point in a $C$-dimensional space, creating a point-cloud that traces out the trajectory of the system through this space as it evolves (*b*). We then computed the persistent homology of the complex defined by these points using cosine distance recording the evolution of connected components and cycles throughout the filtration. The process can be thought of as spheres centred on each embedded point with growing radius. As the radius of the spheres grow and begin to overlap, a discretized approximation of the underlying manifold emerges. To classify the evolving manifold, we use two summaries: the Betti curve, that counts the number of cycles present in the discretized manifold for each filtration radius (*c*); and the persistence diagram, that encodes each cycle as a point and records the radius of its first appearance (birth radius) in the *x*-axis and until which radius it persists in the *y*-axis (*d*).

We construct the embedded point cloud (EPC) by treating each column as a vector in $C$-dimensional space, so at each time stamp the brain state is encoded as a $C$-dimensional vector in the space of all possible states it can assume. As time goes by, we can see the entire recording as a trajectory across this continuous, $C$-dimensional state space. For visualization of this process, see figure 2. The first analysis we did was to calculate the cosine distance between temporally consecutive states. Since the data were recorded with a constant sampling interval, the distance between states at time $t$ and $t + 1$ are proportional to a 'velocity' through the state-space. The average velocity is a measure of how rapidly patterns of activity observed over the array change and the rate at which the system as a whole is evolving.

## 2.3. Topological data analysis

Topology is the area of mathematics that studies shapes and spaces hard to represent visually. More recently, topology has been applied in data analysis to help describe and classify noisy or high-dimensional data, for example by extracting topological invariants present in complex datasets [30,46,47]. Among the many topological methods that have been developed for data analysis, the most frequently used is persistent homology. Persistent homology allows us to build descriptors of the shape of a point cloud, by cataloguing the existence of different structural features, such as connected components, cycles, voids, etc. at different levels of coarse discretization of the data. We can think of the process (known variably as a Rips filtration or a Vietoris–Rips filtration) as spheres centred on each embedded point with growing radius. As the radius of the spheres grows, they begin to intersect, connecting the data points, and a representation of the shape of the data will start to emerge. As one increases the radius, at what scale do we observe changes in representation of the data? To quantify this changing representation, persistent homology takes note of the number of

connected components (zero-dimensional homology) and the radius at which they merge together and when the points in these components start to connect, creating more complex structures like loops (one-dimensional homology), and at what size of the spheres they disappear.

To compare the features, we use Betti curves. For each feature type (connected component and cycles), a Betti curve counts the number of features that exist at each scale in the Rips filtration. Since the curves all have the same support, the increasing radius of the spheres in the embedding space, the curves can be compared between each other or averaged together, finding the radius that creates the most complex feature sets.

## 2.4. Ordinal partition network

Looking at permutation sequences in a time series has been a rising trend in the dynamical systems community in the last 20 years [48,49]. Permutations are a sensitive indicator of the dynamic state of a system and can be efficiently computed, even for long time-series data. One significant benefit of constructing permutations is that it maps a continuous time series to a finite set of discrete permutations, which allows for principled information-theoretic analysis of systems that might not otherwise be amenable, such as permutation entropy [48].

To explore the temporal dynamics of the system, we constructed ordinal partition network (OPN) representations of the data [31,32], following the procedure discussed in [33]. Due to the difficulties associated with multidimensional OPNs [50], we applied this method to every channel in every recording slice individually, and then aggregated the results. This allows us to capture all of the information in each single channel, as multi-channel analysis with OPNs is not technically feasible.

To construct an OPN, begin with a time series $X = x_1, x_2, \ldots, x_n$. This time series is then embedded in a $d$ dimensional space, using a time-lag $\tau$, as is done when constructing a Taken's embedding. The result is a temporally ordered set of vectors of length $d$, where each $v_i = [x_i, x_{i+\tau}, \ldots, x_{i+(d-1)\tau}]$. Each vector $v_i$ is then mapped to the permutation $\pi$, which sorts the coefficients of the vector in increasing order. Sorting the coefficients of $v_i$ we will have that $v_{i,1} \leq v_{i,2} \leq \cdots \leq v_{i,d}$. Each coefficient is then replaced by the position they have in this ordering $\pi$. This new vector will be represented by a permutation of the numbers 1, 2, ..., $dn_i = [\pi(x_i), \pi(x_{i+\tau}), \ldots, \pi(x_{i+(d-1)\tau})]$. We can consider these permutations as nodes in a directed network and connect with a directed edge two permutations that come from consecutive time points. The resulting transition network will have less nodes than existing time points in the original time series, as there might be multiple $i$ for which the respective delay vectors $v_i$ give the same permutation $\pi$. To incorporate this information into the resulting OPN, we give a weight to each node in the network counting how many time points $i$ led to the same permutation (figure 2).

In addition to the topological and information-theoretic measures reported here, we ran a battery of more standard network measures aiming to characterize the connectivity of the OPNs, including measures of centrality (betweenness, Katz), clustering coefficient. For results and discussion, see electronic supplementary material, figure S.I. 1.

### 2.4.1. Free parameters

Like other methods of attractor reconstruction, the OPN algorithm requires two free parameters: the embedding dimension $d$ and the time-lag, $\tau$. There is no agreed-upon method for choosing the optimal $d$ and $\tau$. Different researchers have suggested different criteria, including values that maximize the variance of the degree distribution [32], false-nearest-neighbours criteria [51], or the first local maxima in the permutation entropy [33]. One significant consideration is that the embedding dimension must be large enough that the state-space is sufficiently large to fully capture the range of patterns present in the data (otherwise the resulting OPN fails to capture all the dynamics), but not so large that every moment is unique (in which case, we reconstruct the time series as a path graph). The constructed network should have a sufficiently complex topology so as to represent the richness of the initial dataset, while still finding meaningful patterns. See electronic supplementary material, figure S.I. 5 for an example on the effect of these parameters on the structure of an OPN.

As OPNs are largely restricted to one-dimensional time series, we created a unique OPN for each channel in each of the scans. To account for natural differences in the dynamics of each channel or brain region, each OPN was constructed with a unique embedding lag $\tau$. There were no significant differences between any conditions at the subject level in terms of the average optimal lag. By contrast, every OPN had to be constructed with the same embedding dimension $d$ to enable proper comparison and data aggregation. We selected $\tau$ using the first zero crossing of the autocorrelation,

and $d = 5$ was the mode of the distribution of embedding dimensions for which the variance in the degree distribution was maximal. By constructing networks at the channel level, we can assess changes in brain dynamics at a number of scales, including at the channel level, the region level, and at the individual level by aggregating different numbers of channels. In the Results section, the summary statistics are aggregated at the subject level unless otherwise specified.

### 2.4.2. Information-theoretic analysis of OPNs

A significant benefit of the OPN is that, being a discrete manifold, the state-transition network is amenable to information-theoretic analysis in a way that a continuous manifold is not. Here, we report two measures of how much information a system encodes in its state transition graph [52–54]. The first measure is the determinism, which measures of how much information a system's state transition graph encodes about its future evolution; i.e. how deterministic, on average, is the evolution from state $i$ to state $j$. The determinism is low in a system where each state has an equiprobable chance of evolving into one of many future states, while a system where each state evolves with probability equal to 1 to a subsequent state would be highly deterministic. The average determinism of a directed network $X$ with $N$ can be quantified as

$$\text{Det}(X) = \frac{\log_2(N) - \langle H(W_i^{\text{out}}) \rangle}{\log_2(N)},$$

where $\langle H(W_i^{\text{out}}) \rangle$ corresponds to the average entropy of the probability distribution of possible futures (weighted out-going edges) for each node $i$. The second measure is the degeneracy, which gives a measure of how much information a system's state transition graph encodes about its past evolution. A system where all states feed into the same future would be described as highly degenerate, while a system where each state had a well-defined past would exhibit low degeneracy. Degeneracy is calculated as

$$\text{Deg}(X) = \frac{\log_2(N) - H(\langle W_i^{\text{out}} \rangle)}{\log_2(N)}.$$

These two measures constitute an information-theoretic analysis of state-transition graphs representing a system outputting a continuous signal. The OPN can be thought of as an approximation of the idea of an $\epsilon$-machine [55,56], which provides an optimal approximation of a dynamical system based on output data. Previous work using $\epsilon$-machines to explore the effects of anaesthesia on neural dynamics in insects found that temporal complexity and information asymmetry are strongly altered by loss of consciousness [57], which suggests that these kinds of statistical state-transition analyses can be informative. Combining this information-theoretic formalism provided by Hoel *et al.* [52,54]. with the OPN formalism provides a computationally tractable set of tools to explore the informational structure encoded in continuously varying signals.

## 2.5. Software

All statistical analysis was carried out in Python using the Scipy Stats package (v. 1.1.0) [58,59]. Analysis of variance was computed using the Kruskal–Wallis analysis of variance test [60], and *post hoc* testing was done using the Mann–Whitney $U$ test [61]. Non-parametric statistics were chosen due to the uncertainty that the data were sufficiently Gaussian. OPNs were constructed using the OPyN package (available on Github: https://github.com/thosvarley/OPyN), and the scripts used in this study can be found in the electronic supplementary material. Persistence homology analysis was done using the Ripser (v. 0.3.2) [62] and Persim libraries (v. 0.0.9) as part of the Scikit-TDA library (v. 0.0.4) [63]. Other packages used include the Numpy library (v. 1.15.4) [64], Scikit-Learn (v. 0.20.0) [65], Matplotlib (v. 2.2.2), [66], Spyder (v. 3.2.3), NetworkX (v. 2.2) [67], iGraph (v. 0.7.1) [68]. Analysis was done in the Anaconda Python Environment (Python 3.6.8, Anaconda v. 5.0.0) on Linux Mint 18.3.

# 3. Results

To characterize how propofol and ketamine altered brain dynamics in spatial and temporal domains, we constructed two embedded representations of the data. The first was the embedded point cloud (EPC), which embeds the instantaneous activity across all channels in a high-dimensional space as a point-cloud, which can then be analysed using techniques from topological data analysis. The second

**Table 1.** A table describing all of the measures described here, and how they can be intuitively interpreted. The measures are broadly categorized into several groups, including which embedding they are applied to (EPC versus OPN), and the general mathematical fields they are derived from (EPC, information theory).

| measure | category | formalism | interpretation |
|---|---|---|---|
| maximum persistence | EPC/TDA | the length of the longest lived cycle in the Rips filtration | the degree of 'higher-order' structure present in the EPC |
| total number of cycles | EPC/TDA | the total number of cycles that appears over the Rips filtration | how constrained are the interactions between all channels? |
| maximum number of cycles | EPC/TDA | the maximum number of cycles present at a given moment in the Rips filtration | how constrained are the interactions between all channels? |
| number of nodes | OPN | the number of unique nodes in the OPN | the size of the repetoir of available states |
| number of edges | OPN | the number of unique edges in the OPN | the flexibility with which the system transitions through micro-states |
| determinism | OPN/info. theory | the average entropy of the out-going edges for each node | how predictable is the future given the present? |
| degeneracy | OPN/info. theory | how much information is lost when states run together | how recoverable is the past given the present? |
| modularity | OPN | how well the nodes in the OPN can be clustered | the tendancy for the system to get 'stuck' in smaller subsets of the state space |
| permutation entropy | time series/info. theory | the entropy of the permutation-embedded series | how 'flat' the overall state-space is |
| Lyapunov exponent | time series | the 'chaoticity' of the time series | how predictable is the time series; how sensitive to perturbation |

method, which focuses on temporally extended dynamics, involves constructing a representative state-transition network for each channel (ordinal partition networks, OPNs), which encodes temporally extended dynamics in its structure. We begin by discussing the results of the EPC analysis, and then move on to the OPNs.

## 3.1. Embedded point cloud

For a reference for all of the measures described here, as well as intuitions for what they reveal about brain dynamics, see table 1. Kruskal–Wallis analysis of variance found no significant differences between the three conditions for the maximum persistence of the longest lived cycle (figure 3).

There were significant differences between all the conditions regarding the total number of cycles to exist over the course of the filtration ($H = 62.8$, $p = 2.3 \times 10^{-14}$). The largest total number of cycles was found in the awake condition ($417.5 \pm 119.61$), followed by the ketamine condition ($190.88 \pm 39.24$), and the propofol condition had the fewest number of cycles ($64.17 \pm 19.47$). A similar pattern held for the maximum number of cycles to exist at any individual point in the filtration: Kruskal–Wallis analysis of variance found significant differences between all three conditions ($H = 59.46$, $p = 1.22 \times 10^{-13}$), with the awake condition having the most cycles ($80.0 \pm 29.36$), followed by the ketamine condition ($22.46 \pm 8.09$), and with the propofol condition having the fewest ($10.125 \pm 2.8$). These results suggest that the propofol condition has the least amount of 'structure' constraining the simultaneous evolution of activity across the channels. Recall that, if every channel were acting independently, the resulting EPC would be a smooth, multivariate Gaussian distribution in as many dimensions as there are

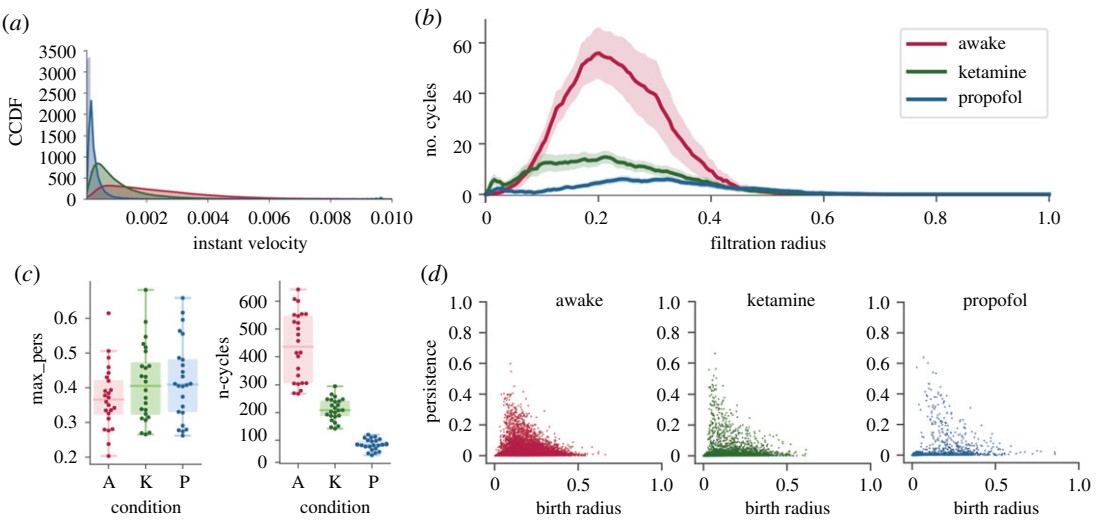

**Figure 3.** Results for the EPC analysis of brain dynamical trajectories in the high-dimension configuration space. Empirical distributions of the instantaneous velocity for the three conditions (*a*). Average Betti curves, with a 90% confidence interval, counting the number of homological cycles present in the complex as the spheres radius increases (*b*). The life duration of the longest cycle in each time segment and total number of homological cycles (represented by a point in the swarm plots), separated by condition (A, awake; K, ketamine; P, propofol) (*c*). Persistence diagrams for all time series embedding separated by conditions. For each condition, a scatter point represents the birth radius and persistence of a cycle in one of the embedding point cloud (*d*).

**Table 2.** Results for the three measures used on the EPC: the maximum persistence of the longest lived cycle, the total number of cycles over the course of the whole filtration, and the maximum number of cycles to exist at any one point in the filtration.

| condition | maximum persistence | number of cycles | maximum number of cycles |
|---|---|---|---|
| awake | 0.36 ± 0.09 | 417.5 ± 119.61 | 80.0 ± 29.36 |
| ketamine | 0.4 ± 0.11 | 190.88 ± 39.24 | 22.46 ± 8.09 |
| propofol | 0.4 ± 0.11 | 64.17 ± 19.47 | 10.125 ± 2.8 |

channels. The existence of cycles suggests a deviation from this maximally entropic ideal—the evolution of the channels appear to be jointly constrained by each other, creating cycles and voids. All results are recorded in table 2 and visualized in figure 4.

## 3.2. Ordinal partition network

We assessed five simple network measures to characterize how the topology of the OPNs changed between conditions. The simplest measure is the number of nodes, which measures the size of the repertoire of ordinal partition micro-states available to the system over the course of it is run. Kruskal–Wallis analysis of variance found significant differences between all three conditions ($H =$ 2296.06, $p < 10^{-20}$), with the awake condition having the most nodes (102.28 ± 28.9), followed by the ketamine condition (86.41 ± 30.14), with the propofol condition having the fewest (65.17 ± 26.6). This is consistent with the original entropic brain hypothesis, that the vividness of consciousness, and complexity of behaviour, tracks the size of the repertoire of available states [19,21].

We also compared the number of edges present in the network. In the same way that the number of nodes counts the unique micro-states the system adopts, the number of edges counts the unique transitions that the system can perform. There were significant differences between all three conditions ($H = 2645.41$, $p < 10^{-10}$, with the awake condition having the most unique edges (480.67 ± 208.95), followed by the ketamine condition (325.13 ± 155.86) and then the propofol condition (191.15 ± 95.46). This suggests that, in addition to the larger repertoire of individual states, there is also increased flexibility in terms of how those states transition between themselves.

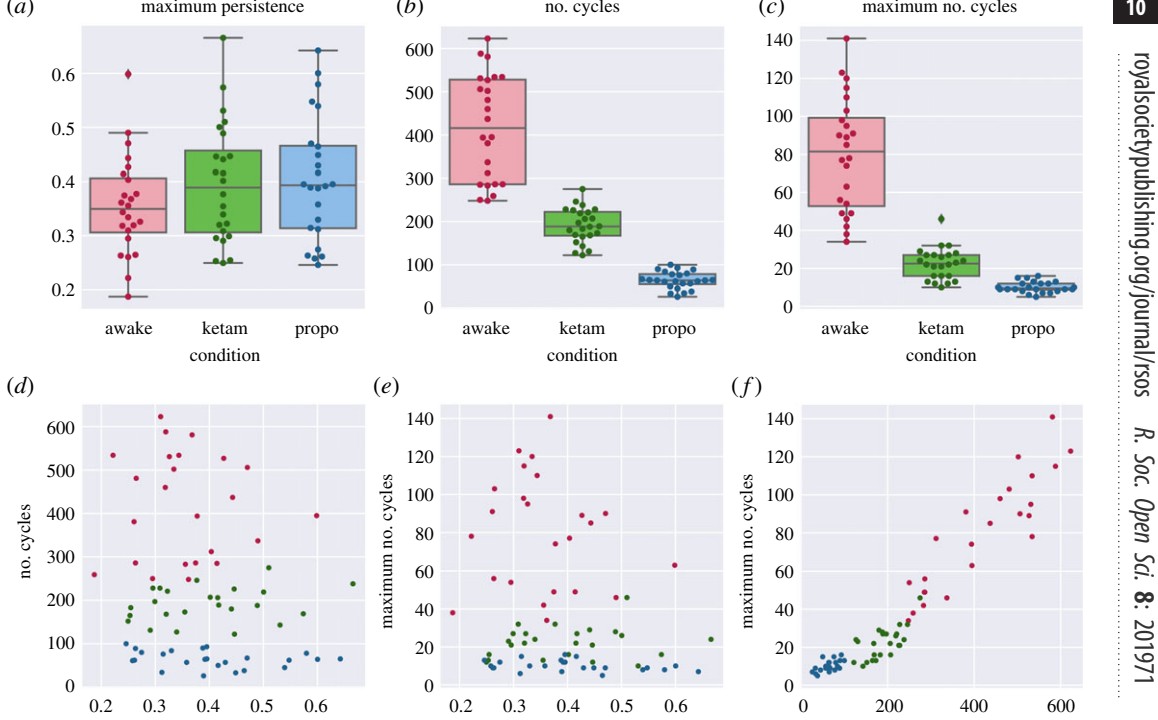

**Figure 4.** (*a,b*) Boxplots showing how the three conditions differed on each of the TDA analyses applied to the EPC: the maximum persistent (*a*), the total number of cycles (*b*), and the maximum number of cycles (*c*). (*d–f*) scatter plots showing how the three measures above relate to each other. Note the strong positive relationship in (*f*), indicating that the total number of cycles is related to the largest number of cycles that appear at a given moment. 'Cycles' in this context can be understood as constraints on the behaviour of the system indicating a deviation from the case where all channels are acting independently.

To leverage the natural application of information theoretic analysis to OPNs, we used two measures originally developed to assess the causal structure of the system: determinism (on average, how predictable is the future from the present) and degeneracy (on average, how well can the past be reconstructed from the present) [54]. There were significant differences between all three conditions ($H = 4678.4$, $p < 10^{-20}$; however, in contrast to previously described measures, it was the propofol condition that had the highest determinism ($0.82 \pm 0.04$ bit), followed by the ketamine condition ($0.86 \pm 0.035$ bit), and the awake condition had the lowest determinism ($0.91 \pm 0.03$ bit). The degeneracy was also significantly different between conditions ($H = 607.94$, $p < 10^{-20}$), with ketamine having the highest degeneracy ($0.02 \pm 0.01$ bit), followed by propofol ($0.0114 \pm 0.009$ bit), and then the awake condition ($0.0112 \pm 0.012$ bit).

The final network measure we applied was the modularity, using the Infomap modularity algorithm [69]. The Infomap algorithm assigns a subset of nodes to the same community if a random walker on the network has a tendency to get 'stuck' in that subset—in the context of a state-transition network, where a random walk is naturally understood as a possible trajectory of the system through state-space, a module could then be understood as a kind of 'metastable attractor' that the system gets transiently caught in. A high modularity, then, is indicative of strong higher-order attractor dynamics constraining the evolution of the system, while a low modularity describes a relatively 'flat' state transition landscape. Kruskal–Wallis analysis of variance found significant differences between all three conditions ($H = 1734.71$, $p < 10^{-20}$), with the propofol condition having the highest modularity ($0.85 \pm 0.13$, followed by the ketamine condition ($0.84 \pm 0.08$), and then the awake condition had the least modular structure ($0.81 \pm 0.08$). All results for this section are tabulated in table 3 and visualized in figure 5.

### 3.2.1. Raw time-series measures

In addition to the analysis of the ordinal partition networks themselves, we performed two classical non-network-based analyses of the time series, to compare how our novel methods compared to more established ones. The first was the permutation entropy [48] (which is intimately related to the

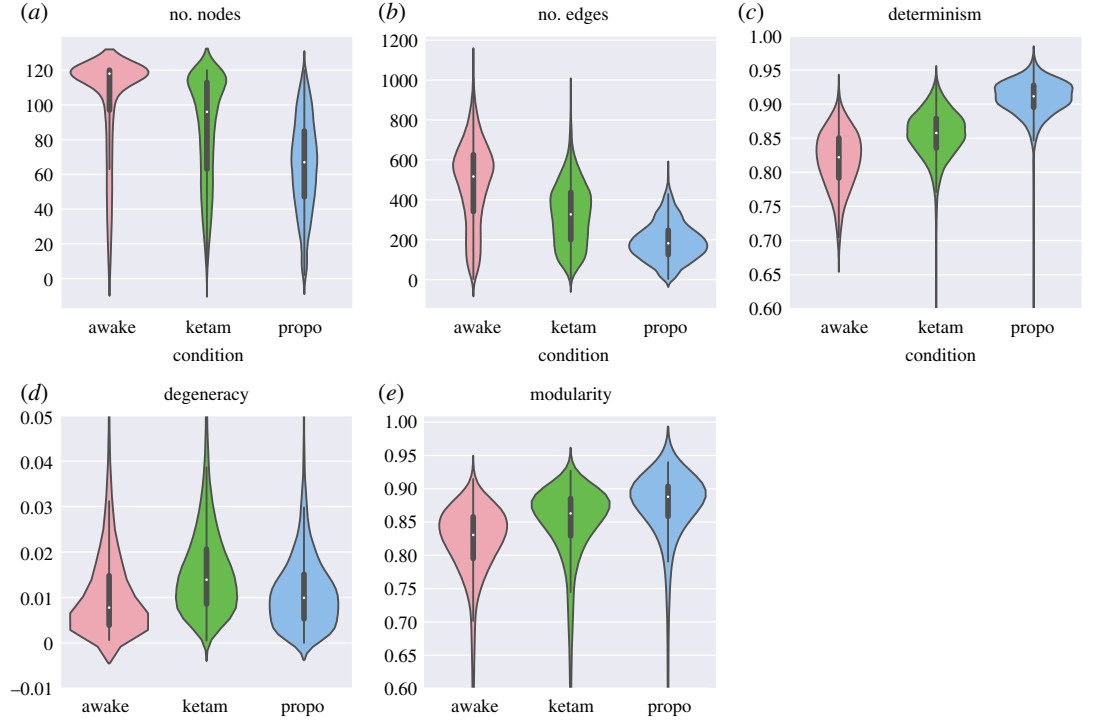

**Figure 5.** Violin plots for the various OPN measures. Violin plots were chosen to represent the distribution of measures over thousands of individual channels. (*a*) The difference between the three conditions in the number of unique nodes present in the OPN. This represents something like a measure of the size of the repertoire of states available to the system in a given condition. (*b*) The number of unique edges in the OPN, represents the flexibility of the systems dynamics. (*c*) The determinism: an information-theoretic measure of how reliably the future of the system can be predicted from its past. (*d*) The degeneracy: a measure of how much information about the past is lost when different states 'run into' each other. This is a rare case where the ketamine condition is higher than either of the awake or propofol conditions. (*e*) The modularity of the OPN network, determined using the Infomap algorithm [69]. Provides a measure of how 'constrained' the systems dynamics are by higher-order patterns of state-transitions.

**Table 3.** Results for the five measures used to characterize the OPNs: the number of nodes in the network, the number of edges, the determinism, degeneracy and modularity of the network. Each of these measures can be thought of as a different axis along which the discrete state-transition dynamics can occur.

| condition | number of nodes | number of edges | determinism | degeneracy | modularity |
|---|---|---|---|---|---|
| awake | 102.28 ± 28.9 | 480.67 ± 208.95 | 0.81 ± 0.04 | 0.0112 ± 0.012 | 0.81 ± 0.08 |
| ketamine | 86.41 ± 30.14 | 325.13 ± 155.86 | 0.86 ± 0.035 | 0.02 ± 0.01 | 0.84 ± 0.08 |
| propofol | 65.17 ± 26.6 | 191.15 ± 95.46 | 0.91 ± 0.03 | 0.0114 ± 0.009 | 0.85 ± 0.13 |

construction of the OPN). Kruskal–Wallis analysis of variance found significant differences between all three conditions (statistic $= 2219.65$, $p < 10^{-20}$). As usual, the awake condition had the highest permutation entropy ($4.27 \pm 0.59$ bit), followed by the ketamine condition ($3.92 \pm 0.58$ bit), and with propofol having the lowest ($3.61 \pm 0.61$). These results indicate that, on average, the system is visiting all of the realized micro-states more equitably in the awake condition, whereas in the propofol condition, it is repeatedly returning to a subset of the states and only rarely visiting others.

The other measure we used was the Lyapunov exponent, commonly understood as a measure of 'how chaotic' a system is. Once again, the usual pattern held: significant differences between all three conditions ($H = 5628.26$, $p < 10^{-20}$), with the awake having the greatest chaoticity ($0.2 \pm 0.013$), following by the ketamine condition ($0.18 \pm 0.01$) and the propofol condition ($0.16 \pm 0.01$). Chaoticity can be thought of as something like a measure of how sensitive a system is to perturbation (how rapidly a perturbed trajectory diverges from it is unperturbed self). This may be naturally understood

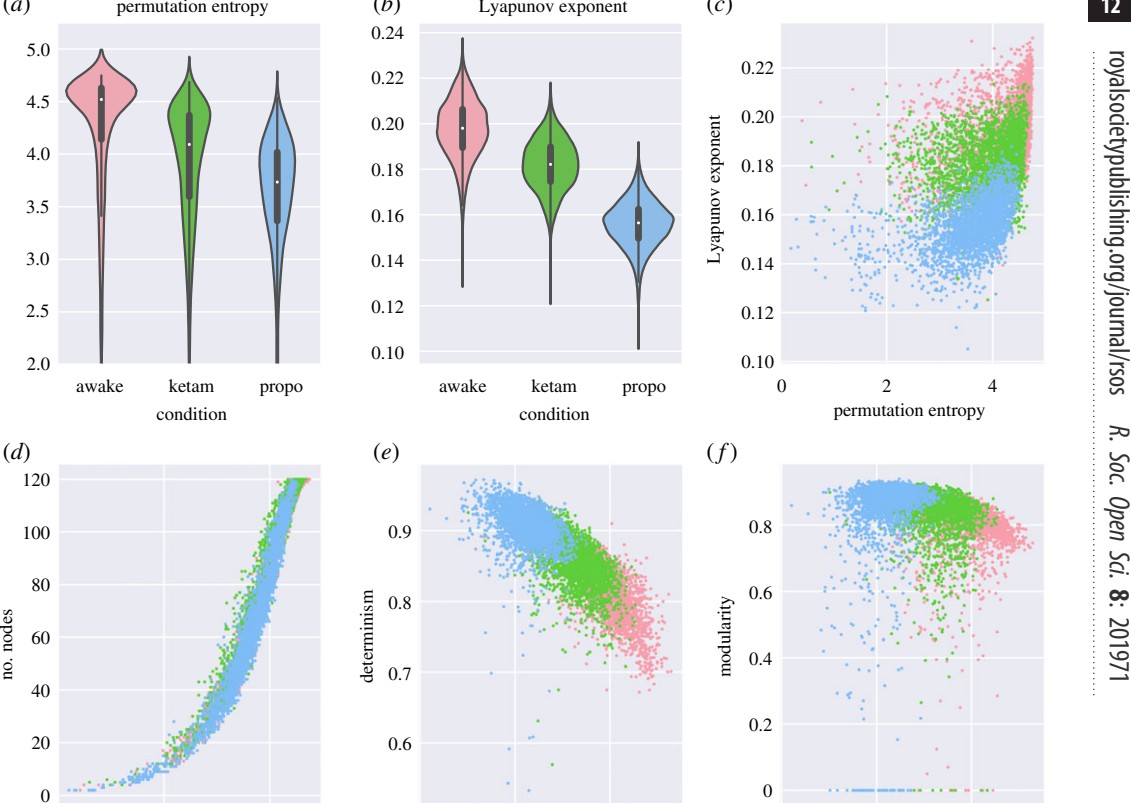

**Figure 6.** (*a,b*) Violin plots of the permutation entropy and the Lyapunov exponent. As with the OPN measures, we chose violin plots to capture the distribution of a large number of channels. (*c*) A scatter plot of the permutation entropy against the Lyapunov exponent: note the positive, but nonlinear relationship between both measures, consistent with previous work showing an association between permutation entropy and chaos [48]. (*d–f*) Scatter plots showing how the two time-series measures relate to the OPN measures. The scatter-plots show that these novel measures derived from the OPN are consistent with established measures. As expected, the number of nodes is positively associated with the permutation entropy (*d*). The Lyapunov exponent is negatively correlated with the determinism (which is consistent with the results reported in [70] and consistent with the intuition behind chaotic systems) (*e*). Finally, there is a slight negative relationship between the Lyapunov exponent and the modularity, suggesting that metastable higher-order dynamics may be harder to maintain in chaotic systems (*f*).

**Table 4.** Results for the two 'classical' measures used to characterize the chaoticity and information content in time series: the permutation entropy and the Lyapunov exponent.

| condition | permutation entropy | Lyapunov exponent |
| --- | --- | --- |
| awake | $4.27 \pm 0.59$ | $0.2 \pm 0.013$ |
| ketamine | $3.92 \pm 0.58$ | $0.18 \pm 0.01$ |
| propofol | $3.61 \pm 0.61$ | $0.16 \pm 0.01$ |

in the context of the need for conscious, awake, animals to be able to rapidly respond to new stimuli from the environment. Sensitivity to environmental perturbations has clear benefits, although in the case of 'overly-chaotic dynamics', it would certainly become detrimental. All results are tabulated in table 4 and visualized in figure 6.

## 3.3. Dimensionality reduction and visualization

We can imagine that each one of the results discussed above defines a kind of 'dynamical morphospace', analogous to previous work that has been done on morphospaces in network topology [71], where every

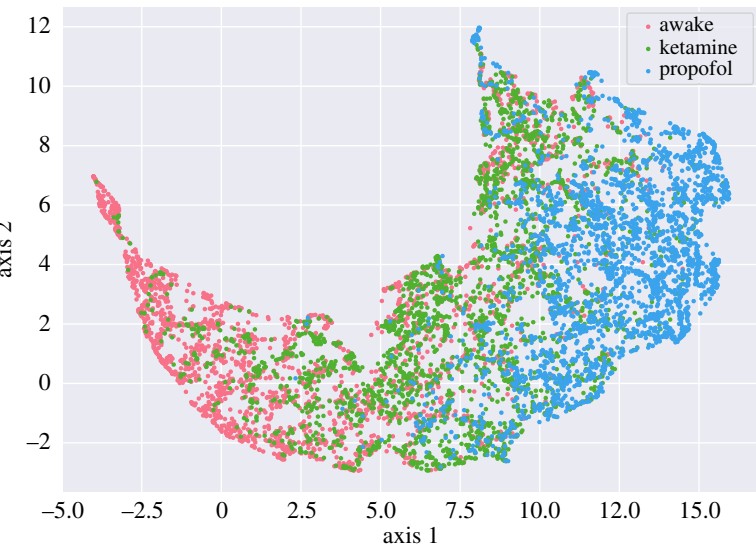

**Figure 7.** The UMAP [72] embedding of the results of the five OPN-based measures for every channel, in each condition. Note that the ketamine condition forms a kind of boundary between the awake and the propofol conditions, with penetration into both regions. This suggests that, when multiple metrics are taken into consideration, the state induced by ketamine combines elements of both normal waking consciousness and propofol anaesthesia. Note that, in a UMAP embedding, the resulting axes are not intrinsically meaningful (in contrast to linear embedding algorithms like PCA): the meaningful information is in the distances between individual points, as opposed to the specific values of the coordinates.

network is embedded in a high-dimensional space: the value along a given axis is defined by the various measures described above (i.e. one axis is determinism, one is degeneracy, etc.). In the case of just the OPN results, this gives us one point for every channel embedded in a five-dimensional morphospace. This can be visualized using a dimensionality reduction algorithm like PCA, tSNE or UMAP to create a 'birds-eye-view' of how the different states of consciousness relate to each other. While this does not return a quantitative measure of similarity or difference, it provides a useful visualization of how the different conditions are related to each other, which can be more intuitive than a table of numbers. We used the UMAP embedding algorithm [72] to construct a visualization of the channel-wise relationships between the conditions using just the OPN data (figure 7).

## 4. Discussion

In this paper, we have discussed several ways in which temporal and spatial embedding of electrophysiological data from macaques in three distinct states of consciousness (awake, ketamine anaesthesia, propofol anaesthesia) can reveal insights into how brain dynamics reflect alterations to consciousness along many axes. Historically, there has been considerable interest in one-dimensional, scalar measures of how the 'complexity' of brain activity relates to consciousness (e.g. Lampel–Ziv complexity [3,17] or integrated information theory's 'Phi' value [36]). However, as argued by Feldman & Crutchfield [73], there are fundamental limitations to how much insight can be gained by even a 'well-behaved' scalar measure of complexity. This notion was recently given empirical support by the finding that, when attempting to discriminate between conscious and anaesthetized states, high-dimensional information structures did a significantly better job than standard scalar measures [74]. This strongly suggests that when attempting to characterize a system as complex as a conscious (or even unconscious) brain, which can vary along many different axes, a more comprehensive picture is necessary. By constructing two embeddings (the EPC and the channel-wise OPNs), we can paint a much more holistic picture of how brain activity changes in spatial and temporal domains when consciousness is lost or altered. Rather than creating a ranking of 'complexity' from low to high, we can begin to tease out the ways in which these conditions are similar, and different.

The awake condition could be characterized as having a high degree of interaction between the individual channels when compared with propofol, as evidence by the persistence homology results: the presence of a large number of cycles suggests that the dynamics of the individual regions are

subject to collective constraints that are less prominent when consciousness is lost. At the level of individual channels, the awake condition can adopt the largest number of unique micro-states and has the highest degree of flexibility transitioning between them. At the macro-scale, modularity analysis revealed that the state-transition network has the lowest modular structure, which indicates that the system is less likely to get caught in deep attractors compared to the propofol condition. Finally, the awake condition is both less deterministic and less degenerate, which previous work has found to be indicative of the onset of chaotic dynamics [70], an interpretation supported by the Lyapunov exponents as well. This suggests that the temporal dynamics of the awake condition are the least predictable, suggesting a high-degree of flexibility compared to either anaesthesia states.

We can compare these findings with the results from the propofol analysis. Persistence homology analysis of the EPC found that the propofol condition had the lowest number of cycles, suggesting a loss of 'higher-order structure' driving activity across multiple channels. This is consistent with previous findings that propofol anaesthesia decreases functional connectivity [75]; however, one significant benefit of the persistence homology analysis is that it considers the joint-states of all channels together, as opposed to examining pairwise relationships between individual channels, which may miss higher-order synergies that may be present in the system [76,77]. The propofol condition had the smallest repertoire of available states and comparatively constrained transitions between them. Modularity analysis bore this out, finding that the propofol condition had a significantly higher modularity, suggesting that the system is more likely to get 'stuck' in subsets of the state space. It was also the most deterministic, suggesting reduced dynamical flexibility. Interestingly, it was the ketamine condition that had the most degenerate dynamics, suggesting that the ketamine has the shortest 'memory', as the past states are minimally predictable from the present.

In general, the ketamine condition occupied something of a middle ground between the awake and propofol conditions, suggesting that it combines elements of both states in its dynamics. This is clearly visible when the UMAP embedding is performed on the OPN-morphospace: the ketamine condition is clearly visible forming a kind of boundary between the awake and propofol conditions, which do not overlap significantly. This is consistent with the known clinical properties of ketamine anaesthesia: while it produces a state that is externally very similar to propofol anaesthesia (loss of responsiveness to stimuli, analgesia, etc.), ketamine can produce dream-like, dissociative states [2,16], suggesting that the process generating phenomonological consciousness is not completely inhibited.

This work does have limitations which are worth considering. The most obvious is the small sample size: two macaques is a small N, even with multiple slices taken out of the longer scans. Given the origin of this data, this limitation cannot be currently addressed and we hope that these results, and the larger methods introduced, can replicate these findings in future studies. We also cannot directly infer what state of consciousness the macaques were in at any given time, or even if macaques are capable of experiencing something like the dissociative anaesthesia that ketamine induces in humans. As with the small N, this is something of a fundamental limitation and an ongoing issue in consciousness research. In terms of the OPNs, a significant limitation is that they can only be constructed from a single channel: while attempts at multivariate generalizations have been proposed [50], 128-channel systems such as those explored here remain computationally and practically prohibitive. This highlights the importance of multiple different measures to bear on a question, as opposed to looking for a singular test that explains 'everything.' The OPN and EPC framework may be complemented by other research frameworks that explicitly aim to understand 'integration' in the form of statistical dependencies between many interacting elements of the brain, for example, the recent work on consciousness and integrated information decomposition (ΦID) [78,79], consciousness and critical brain dynamic [80,81], functional connectivity network analysis [12,82,83] and integrated information theory [36,84]. Given that previous research suggests that this kind of 'integration' is key for the maintenance of consciousness, a key future refinement of the topological data analysis framework would be incorporating measures of integration and higher-order statistical dependencies. Within the time-delay and state-space reconstruction framework, work on cross-embeddings using the same Neurotycho data has found that multivariate state-space reconstruction can yield insights into how anaesthesia changes the interactions between brain regions [85]. This approach could be unified with approaches for constructing cross- and joint-OPNs [86] to enable the applications of our methods to multivariate datasets.

The work presented here is explicitly data driven, rather than theory driven. There are a large number of competing theories of consciousness, such as IIT [36], the information closure theory of consciousness [87], and the global workspace theory [88], to name a few, and rather than attempting to adjudicate between them, we instead developed these analytical pipelines to empower future researchers

interested in empirically testing the various theories of consciousness. We anticipate that the techniques described here can be used to understand other states of consciousness, such as psychedelia or disorders of consciousness following brain injury, as well as understanding individual differences in normal cognition. We might hypothesize, for instance, that high performance on creativity tasks might be associated with an increase in the repertoire of micro-states discernible by the OPN and the flexibility with which the brain transitions between them. Using the notion of a dynamical morphospace, it may be possible to create a 'map' of different cognitive processes based on their dynamical similarities and differences.

# 5. Conclusion

In this work, we describe how embeddings of neural activity data can help differentiate between the similarities and differences between three distinct states of consciousness: normal waking awareness, propofol anaesthesia and ketamine anaesthesia. To assess the spatial distribution of activity across channels, we used topological data analysis to analyse the structure of the joint-states of all channels through time. To assess the channel-level temporal dynamics, we construct discrete state-transition graphs using ordinal partition networks, which reveal how the system evolves through state-space in time. We found that the awake condition was characterized by both a high-degree of inter-channel interactions, as well as a more flexible, less predictable structure, in contrast to propofol which had less inter-channel interaction, and more predictable, constrained dynamics. Ketamine anaesthesia sat between the two extremes, combining elements of both. By combining multiple measures into a sort of 'dynamical morphospace', we can better understand how distinct states relate to each other.

Data accessibility. This article has no additional data.

Authors' contributions. A.P. and T.F.V. conceptualized and designed the experiment. A.P., T.F.V. and V.D. performed data analysis. A.P., T.F.V. and O.S. made figures, wrote and edited the paper.

Competing interests. We declare we have no competing interests.

Funding. T.F.V. is supported by NSF-NRT grant no. 1735095, Interdisciplinary Training in Complex Networks and Systems. A.P. is supported by the Indiana University Network Science Institute (IUNI). T.F.V. and O.S. are supported by NIH/NIMH grant no. 1RO1MH121978-01.

Acknowledgements. We thank Joshua Faskowitz for his wit and wisdom throughout the process.

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
