## [Peer Review File · Royal Society Open Science]

Review History

RSOS-201971.R0 (Original submission)

Review form: Reviewer 1 (Nao Tsuchiya)

Is the manuscript scientifically sound in its present form?

Yes

Are the interpretations and conclusions justified by the results?

Yes

Is the language acceptable?

Yes

Do you have any ethical concerns with this paper?

No

Have you any concerns about statistical analyses in this paper?

No

Recommendation?

Accepted with minor revision (please list in comments)

Comments to the Author(s)

This paper presents two analyses on the previously collected open data (NeuroTycho). Their first analysis uses EPC with TDA and the second uses OPNs. I have some comments to improve the paper, but I think this paper can be accepted with minor revision.

Signed review by Naotsugu Tsuchiya.

Major issues.

1. Consideration of “nonconscious neural activity” vs. high dimensional space of neural activity

While you mention the issue of conscious vs. nonconscious neural activity in some places, I would suggest that you discuss this issue in Discussion as this is a serious issue in consciousness research.

The paper reviews an adequate amount of the literature on the “level” of consciousness, however, it doesn’t cover “contents” of consciousness. This is fine in itself. However, the latter literature has generated a huge amount of empirical evidence to suggest that not all neural activities are relevant for consciousness (See for example, Koch 2016 Nat Rev Neuro, Mashour et al 2020 Neuron). Given this, including all the available channels (>120) for the analysis can possibly reflect neural activities that are irrelevant for supporting consciousness but more directly related to nonconscious processing. This needs to be acknowledged and you may want to discuss how to resolve this issue in the future.

One possible approach is the identification of “complex” as suggested by the integrated information theory (IIT) by Tononi.

2. Structural measures/characterizations of consciousness

Page 3. L15 -

“these point-summary measures, while informative, collapse multi-scale dynamics into a single number and thus have difficulty capturing its specific shape or form.”

I totally agree with this statement. And I would say that this is pretty much in line with the philosophy of the integrated information theory (IIT). This aspect has become more explicit since IIT3.0. Recently, we have published a paper directly addressing this structural and topological consideration on the level of consciousness based on the empirical neural data (Leung et al 2021 PLoS Comp), which you might want to look at.

Also, as you refer to epsilon machines several times in the manuscript as an alternative way to characterize the topology/shape of information structure, it may be worth looking at our recent paper where we applied the epsilon machine on the loss of consciousness (Munos et al 2020 Physical Review Research).

3. Limits of OPN

With respect to the limit of OPN, I think it is better to mention other approaches that have also taken multivariate approaches. IIT can be considered as one of them, where it explicitly deals with the integration among channels, which your OPN approach explicitly ignores. To the extent

that integration is critical to understand consciousness, this may be a potential oversight / limitation of your approach.

Along with this line, I also think you should acknowledge a couple of papers that have already analyzed the same NeuroTycho data with different ways (e.g., Tajima et al 2015 PLoS Comp).

4. Lack of principles and theories.

Your approach is data-driven and highly descriptive. It may make more sense to explicitly admit this in Discussion and discuss its limitations. Alternatively, you may want to try to link your findings with predictions from some theories. I would imagine some of the findings can be linked with the other theories of consciousness other than the Entropy hypothesis or IIT, but I will leave this to you.

Minor issues

Page 2, line 30-

Lots of double negatives make this paragraph difficult to read for those who are not familiar with these concepts.

Line 34: "light central nervous system stimulation". Probably, you meant that weak stimulation of the nervous system? Better if you can revise here.

Line 41: result it -> result in?

P3 L5: Ref 51&53 didn't really measure entropy. The sentence needs to be revised.

P4 L34: Description of Chibi doesn't make sense. Probably typo for George?

P8: Figure 2 mentions "temporal principal component" but this is not explained anywhere else in the manuscript. (By the way, I was not able to access to the Supplementary Material for this paper)

P8: No significant differences on the lag parameter is mentioned but the data is not shown. Please show it on Supplementary Material.

P8: "a directed network X with N" -> "a directed network X with N nodes/states"?

P9: the equations for determinism and degeneracy are identical. I think the one for degeneracy should use W_{in} ?

P10: The right side of Table 1 is not visible (if something is there?)

P13: Determinism (Fig 5C) is lowest for awake, which is inconsistent with Table 3. I guess the figure is correct?

There are several typos throughout. (e.g., "spacial" "repetoire" "asses")

Decision letter (RSOS-201971.R0)

Dear Mr Varley

On behalf of the Editors, we are pleased to inform you that your Manuscript RSOS-201971 "Topological Analysis of Differential Effects of Ketamine and Propofol Anesthesia on Brain Dynamics" has been accepted for publication in Royal Society Open Science subject to minor revision in accordance with the referees' reports. Please find the referees' comments along with any feedback from the Editors below my signature.

Please submit your revised manuscript and required files (see below) no later than 7 days from today's (ie 19-Apr-2021) date. Note: the ScholarOne system will 'lock' if submission of the revision is attempted 7 or more days after the deadline. If you do not think you will be able to meet this deadline please contact the editorial office immediately.

on behalf of Dr Mark Walton (Associate Editor) and Essi Viding (Subject Editor)
openscience@royalsociety.org

Associate Editor Comments to Author (Dr Mark Walton):

I'm sorry that it has taken so long to get the review back to you. It proved unprecedentedly difficult to secure reviewers for this article, in spite of the best efforts of all. Given the nature of the comments, I will be happy to make a final decision on a revised version without sending it out to review again.

Reviewer comments to Author:

Reviewer: 1
Comments to the Author(s)

This paper presents two analyses on the previously collected open data (NeuroTycho). Their first analysis uses EPC with TDA and the second uses OPNs. I have some comments to improve the paper, but I think this paper can be accepted with minor revision.

Signed review by Naotsugu Tsuchiya.

Major issues.

1. Consideration of “nonconscious neural activity” vs. high dimensional space of neural activity

While you mention the issue of conscious vs. nonconscious neural activity in some places, I would suggest that you discuss this issue in Discussion as this is a serious issue in consciousness research.

The paper reviews an adequate amount of the literature on the “level” of consciousness, however, it doesn’t cover “contents” of consciousness. This is fine in itself. However, the latter literature has generated a huge amount of empirical evidence to suggest that not all neural activities are relevant for consciousness (See for example, Koch 2016 Nat Rev Neuro, Mashour et al 2020 Neuron). Given this, including all the available channels (>120) for the analysis can possibly reflect neural activities that are irrelevant for supporting consciousness but more directly related to nonconscious processing. This needs to be acknowledged and you may want to discuss how to resolve this issue in the future.

One possible approach is the identification of “complex” as suggested by the integrated information theory (IIT) by Tononi.

2. Structural measures/characterizations of consciousness

Page 3. L15 -

“these point-summary measures, while informative, collapse multi-scale dynamics into a single number and thus have difficulty capturing its specific shape or form.”

I totally agree with this statement. And I would say that this is pretty much in line with the philosophy of the integrated information theory (IIT). This aspect has become more explicit since IIT3.0. Recently, we have published a paper directly addressing this structural and topological consideration on the level of consciousness based on the empirical neural data (Leung et al 2021 PLoS Comp), which you might want to look at.

Also, as you refer to epsilon machines several times in the manuscript as an alternative way to characterize the topology/shape of information structure, it may be worth looking at our recent paper where we applied the epsilon machine on the loss of consciousness (Munos et al 2020 Physical Review Research).

3. Limits of OPN

With respect to the limit of OPN, I think it is better to mention other approaches that have also taken multivariate approaches. IIT can be considered as one of them, where it explicitly deals with the integration among channels, which your OPN approach explicitly ignores. To the extent that integration is critical to understand consciousness, this may be a potential oversight / limitation of your approach.

Along with this line, I also think you should acknowledge a couple of papers that have already analyzed the same NeuroTycho data with different ways (e.g., Tajima et al 2015 PLoS Comp).

4. Lack of principles and theories.

Your approach is data-driven and highly descriptive. It may make more sense to explicitly admit this in Discussion and discuss its limitations. Alternatively, you may want to try to link your

findings with predictions from some theories. I would imagine some of the findings can be linked with the other theories of consciousness other than the Entropy hypothesis or IIT, but I will leave this to you.

Minor issues

Page 2, line 30-

Lots of double negatives make this paragraph difficult to read for those who are not familiar with these concepts.

Line 34: "light central nervous system stimulation". Probably, you meant that weak stimulation of the nervous system? Better if you can revise here.

Line 41: result it -> result in?

P3 L5: Ref 51&53 didn't really measure entropy. The sentence needs to be revised.

P4 L34: Description of Chibi doesn't make sense. Probably typo for George?

P8: Figure 2 mentions "temporal principal component" but this is not explained anywhere else in the manuscript. (By the way, I was not able to access to the Supplementary Material for this paper)

P8: No significant differences on the lag parameter is mentioned but the data is not shown. Please show it on Supplementary Material.

P8: "a directed network X with N" -> "a directed network X with N nodes/states"?

P9: the equations for determinism and degeneracy are identical. I think the one for degeneracy should use W_{in} ?

P10: The right side of Table 1 is not visible (if something is there?)

P13: Determinism (Fig 5C) is lowest for awake, which is inconsistent with Table 3. I guess the figure is correct?

There are several typos throughout. (e.g., "spacial" "repetoire" "asses")

===PREPARING YOUR MANUSCRIPT===

===PREPARING YOUR REVISION IN SCHOLARONE===

Author's Response to Decision Letter for (RSOS-201971.R0)

See Appendix A.

Decision letter (RSOS-201971.R1)

Dear Mr Varley,

It is a pleasure to accept your manuscript entitled "Topological Analysis of Differential Effects of Ketamine and Propofol Anesthesia on Brain Dynamics" in its current form for publication in Royal Society Open Science.

Please see the Royal Society Publishing guidance on how you may share your accepted author manuscript at <https://royalsociety.org/journals/ethics-policies/media-embargo/>. After publication, some additional ways to effectively promote your article can also be found here

<https://royalsociety.org/blog/2020/07/promoting-your-latest-paper-and-tracking-your-results/>.

on behalf of Dr Mark Walton (Associate Editor) and Essi Viding (Subject Editor)
openscience@royalsociety.org

Appendix A

1 **Response to Reviewers**

2 Reviewer: 1

3 Comments to the Author(s)

4 This paper presents two analyses on the previously collected open data (NeuroTycho). Their
5 first analysis uses EPC with TDA and the second uses OPNs. I have some comments to improve
6 the paper, but I think this paper can be accepted with minor revision.

7 Signed review by Naotsugu Tsuchiya.

8 We would like to thank Dr. Tsuchiya for his thoughtful and insightful comments on this paper.
9 We have responded to them below and hope that they are satisfactory. Our responses are noted in
10 blue font. Where we are quoting from the main manuscript, we use *italicized font*.

11 - Thomas Varley (on behalf of all authors)

12 **Major issues.**

13 1. Consideration of “nonconscious neural activity” vs. high dimensional space of neural activity

14 While you mention the issue of conscious vs. nonconscious neural activity in some places, I would
15 suggest that you discuss this issue in Discussion as this is a serious issue in consciousness research.

16 The paper reviews an adequate amount of the literature on the “level” of consciousness, however,
17 it doesn’t cover “contents” of consciousness. This is fine in itself. However, the latter literature has
18 generated a huge amount of empirical evidence to suggest that not all neural activities are relevant
19 for consciousness (See for example, Koch 2016 Nat Rev Neuro, Mashour et al 2020 Neuron). Given
20 this, including all the available channels (≈ 120) for the analysis can possibly reflect neural activities
21 that are irrelevant for supporting consciousness but more directly related to nonconscious processing.
22 This needs to be acknowledged and you may want to discuss how to resolve this issue in the future.

23 One possible approach is the identification of “complex” as suggested by the integrated informa-
24 tion theory (IIT) by Tononi.

25 The distinction between conscious and non-conscious complex activity is a good one - we agree
26 that it is worth discussing. We were, however, limited in our ability to make any inferences about
27 the content of consciousness at all since 1. anaesthesia is typically light on content and 2. the animal
28 models cannot report their experience. We have added the following to the Introduction to make
29 the distinction between level and content of consciousness clear:

30 *We should note that in this project we have focused primarily on the issue of level of consciousness
31 rather than the content of consciousness. This is a subtle distinction that has been discussed in
32 detail (for review, see Koch, 2016) but briefly, the level of consciousness quantifies the “amount”
33 of consciousness, such as the vividness or intensity of subjective experience, while the content of
34 consciousness refers to the specific perceptions that are being consciously perceived. The question
35 of the content of consciousness is well explored by psychophysical studies (Wackerman, 2010) and
36 more recently discussed theoretically in the context of Integrated Information Theory (Tononi, 2008,
37 Oizumi et al., 2014) however it is beyond the focus of the results presented here, for several reasons.
38 Primarily, anaesthetic states are typically light on complex contents, and macaques are unable to*

39 *report their subjective experience, we have no access to the contents of their consciousness, only their*
40 *status as awake or anesthetized based on externally observable variables, and the drug in question.*

41 2. Structural measures/characterizations of consciousness

42 Page 3. L15 - “these point-summary measures, while informative, collapse multi-scale dynamics
43 into a single number and thus have difficulty capturing its specific shape or form.”

44 I totally agree with this statement. And I would say that this is pretty much in line with the
45 philosophy of the integrated information theory (IIT). This aspect has become more explicit since
46 IIT3.0. Recently, we have published a paper directly addressing this structural and topological
47 consideration on the level of consciousness based on the empirical neural data (Leung et al 2021
48 PLoS Comp), which you might want to look at.

49 We have added a reference to the work by Leung et al., which is a fascinating piece of research.

50 *This notion was recently given empirical support by the finding that, when attempting to dis-*
51 *criminate between conscious and anesthetized states, high-dimensional information structures did a*
52 *significantly better job than standard scalar measures (Leung et al., 2021). This strongly suggests*
53 *that when attempting to characterize a system as complex as a conscious (or even unconscious)*
54 *brain, which can vary along many different axes, a more comprehensive picture is necessary*

55 Also, as you refer to epsilon machines several times in the manuscript as an alternative way to
56 characterize the topology/shape of information structure, it may be worth looking at our recent
57 paper where we applied the epsilon machine on the loss of consciousness (Munos et al 2020 Physical
58 Review Research).

59 I added the Munoz citation and discussed it briefly in the context of ϵ -machines. *Previous*
60 *work using ϵ -machines to explore the effects of anaesthesia on neural dynamics in insects found*
61 *that temporal complexity, and information asymmetry are strongly altered by loss of consciousness*
62 *(Munoz et al., 2020), which suggests that these kinds of statistical state-transition analyses can be*
63 *informative.*

64 3. Limits of OPN

65 With respect to the limit of OPN, I think it is better to mention other approaches that have also
66 taken multivariate approaches. IIT can be considered as one of them, where it explicitly deals with
67 the integration among channels, which your OPN approach explicitly ignores. To the extent that
68 integration is critical to understand consciousness, this may be a potential oversight / limitation of
69 your approach.

70 We have expanded the discussion to include references to several different frameworks that explic-
71 itly explore integration between many elements, including the recent work in integrated information
72 decomposition, historical work on functional connectivity network approaches, IIT, and work on
73 criticality and consciousness. We stress that the methods developed here may be useful additions to
74 existing frameworks, rather than replacements.

75 *The OPN and EPC framework may be complemented by other research frameworks that explic-*
76 *itly aim to understand “integration” in the form of statistical dependencies between many interacting*
77 *elements of the brain, for example the recent work on consciousness an integrated information decom-*
78 *position (ΦID) (Luppi et al., 2020a, Luppi et al., 2020b), consciousness and critical brain dynamic*
79 *(Fekete et al., 2018, Varley et al., 2020) functional connectivity network analysis (Lewis et al.,*

2012, Demertzi et al., 2019, Cavanna et al., 20128), and integrated information theory (Tononi, 2008, Toker et al., 2019). Given that previous research suggests that this kind of “integration” is key for the maintenance of consciousness a key future refinement of the topological data analysis framework would be incorporating measures of integration and higher-order statistical dependencies.

Along with this line, I also think you should acknowledge a couple of papers that have already analyzed the same NeuroTycho data with different ways (e.g., Tajima et al 2015 PLoS Comp).

We have added the following to the discussion:

Within the time-delay and state-space reconstruction framework, work on cross-embeddings using the same NeuroTycho data has found that multivariate state-space reconstruction can yield insights into how anaesthesia changes the interactions between brain regions (Tajima et al., 2015). This approach could be unified with approaches for constructing cross- and joint-OPNs (Guo et al., 2018) to enable the applications of our methods to multivariate datasets.

4. Lack of principles and theories.

Your approach is data-driven and highly descriptive. It may make more sense to explicitly admit this in Discussion and discuss its limitations. Alternatively, you may want to try to link your findings with predictions from some theories. I would imagine some of the findings can be linked with the other theories of consciousness other than the Entropy hypothesis or IIT, but I will leave this to you.

Done - we make it explicit that we are doing a data-driven analysis and deliberate choose not to adjudicate between various theories of consciousness.

he work presented here is explicitly data driven, rather than theory-driven. There are a large number of competing theories of consciousness, such as IIT (Tononi, 2008), the information closure theory of consciousness (Chang et al., 2020), and the global workspace theory (Mashour, 2020), to name a few, and rather than attempting to adjudicate between them, we instead developed these analytical pipelines to empower future researchers interested in empirically testing the various theories of consciousness.

Minor issues

Page 2, line 30- Lots of double negatives make this paragraph difficult to read for those who are not familiar with these concepts.

I can't seem to find what he's referring to. Perhaps I am too familiar with these concepts

Line 34: “light central nervous system stimulation”. Probably, you meant that weak stimulation of the nervous system? Better if you can revise here.

Fixed

Line 41: result it -_j result in?

Fixed

P3 L5: Ref 51&53 didn't really measure entropy. The sentence needs to be revised.

I'm not sure what this is referring to here - we don't cite 51 and 53 on page 3?

P4 L34: Description of Chibi doesn't make sense. Probably typo for George?

I am not entirely clear what this refers to.

119 P8: Figure 2 mentions “temporal principal component” but this is not explained anywhere else
120 in the manuscript. (By the way, I was not able to access to the Supplementary Material for this
121 paper)

122 This was a typo left in from an earlier iteration of the project - there were no temporal PCs
123 involved in this analysis.

124 P8: No significant differences on the lag parameter is mentioned but the data is not shown.
125 Please show it on Supplementary Material.

126 These data have been added to the S.I. in the form of a figure

127 P8: “a directed network X with N” -¿ “a directed network X with N nodes/states”? P9: the
128 equations for determinism and degeneracy are identical. I think the one for degeneracy should use
129 W_{in} ?

130 They equations are not the same, although they look very similar. The determinism has the
131 term $\langle H(W^{out}) \rangle$ in it (the average entropy of all the rows of the TPM, while the degeneracy has the
132 term $H(\langle W^{out} \rangle)$ which is the entropy of the average row.

133 P10: The right side of Table 1 is not visible (if something is there?)

134 The right side is a tad cut off due to the LaTeX formatting, but there are no missing columns.
135 This will presumably be fixed when the paper is formatted for the journal.

136 P13: Determinism (Fig 5C) is lowest for awake, which is inconsistent with Table 3. I guess the
137 figure is correct?

138 Yes, I goofed up the LaTeX table. Thank you for catching that - the table has been corrected.

139 There are several typos throughout. (e.g., “spacial” “repetoire” “asses”) Thank you for the
140 detailed proofreading